# Spatiotemporal variations in retrovirus-host interactions among Darwin's finches

Jason Hill [1,6] ✉, Mette Lillie [1,6] ✉, Mats E. Pettersson [1], Carl-Johan Rubin [1,2], B. Rosemary Grant[3], Peter R. Grant[3], Leif Andersson [1,4,5] & Patric Jern [1] ✉

Endogenous retroviruses (ERVs) are inherited remnants of retroviruses that colonized host germline over millions of years, providing a sampling of retroviral diversity across time. Here, we utilize the strength of Darwin's finches, a system synonymous with evolutionary studies, for investigating ERV history, revealing recent retrovirus-host interactions in natural populations. By mapping ERV variation across all species of Darwin's finches and comparing with outgroup species, we highlight geographical and historical patterns of retrovirus-host occurrence, utilizing the system for evaluating the extent and timing of retroviral activity in hosts undergoing adaptive radiation and colonization of new environments. We find shared ERVs among all samples indicating retrovirus-host associations pre-dating host speciation, as well as considerable ERV variation across populations of the entire Darwin's finches' radiation. Unexpected ERV variation in finch species on different islands suggests historical changes in gene flow and selection. Non-random distribution of ERVs along and between chromosomes, and across finch species, suggests association between ERV accumulation and the rapid speciation of Darwin's finches.

Retroviruses represent a diverse group of RNA viruses that must convert their genomes to proviral DNA and integrate permanently in the host nuclear DNA, in order to produce virus progeny. Retroviruses have infiltrated vertebrate host genomes over millions of years via sporadic insertions into the germline, which are inherited as endogenous retroviruses (ERVs)[1,2]. Each ERV is a sample from the retrovirus diversity at the time of germline invasion and evolves largely at the same rate as the host genome, providing a traceable record of long-term retrovirus-host interactions and evolution[1,2]. Many ERVs are ancient and shared by extant descendant host species, thus demonstrating that retroviral infections can impart a genomic legacy shared between species, which is essential for understanding host biology and evolution[1,2].

Although most ERVs are eroded by mutations or the loss of proviral domains through recombination between their long terminal repeats (LTRs), their effects on host genome structure and function are diverse. For example, ERVs can alter the host transcriptome by providing novel genes, by introducing regulatory elements for adjacent host gene expression, or by facilitating recombination that shuffles genomic sequences into new contexts, sometimes with considerable impact on the evolutionary history of their hosts[1,2]. Here, we show that by combining a deeply studied system of natural host populations with a thorough screening of ERVs, we can evaluate aspects of the natural history of the entire Darwin's finches' radiation and retrovirus diversity derived from associated ERVs that would otherwise be elusive.

[1]Science for Life Laboratory, Department of Medical Biochemistry and Microbiology, Uppsala University, SE-751 23 Uppsala, Sweden. [2]Institute of Marine Research, P.O. Box 1870Nordnes NO-5817 Bergen, Norway. [3]Department of Ecology & Evolutionary Biology, Princeton University, Princeton, NJ 08544, USA. [4]Department of Animal Breeding and Genetics, Swedish University of Agricultural Sciences, SE-750 07 Uppsala, Sweden. [5]Department of Veterinary Integrative Biosciences, Texas A&M University, College Station, TX 77843, USA. [6]These authors contributed equally: Jason Hill, Mette Lillie. ✉e-mail: jason.hill@imbim.uu.se; mette.lillie@imbim.uu.se; Patric.Jern@imbim.uu.se

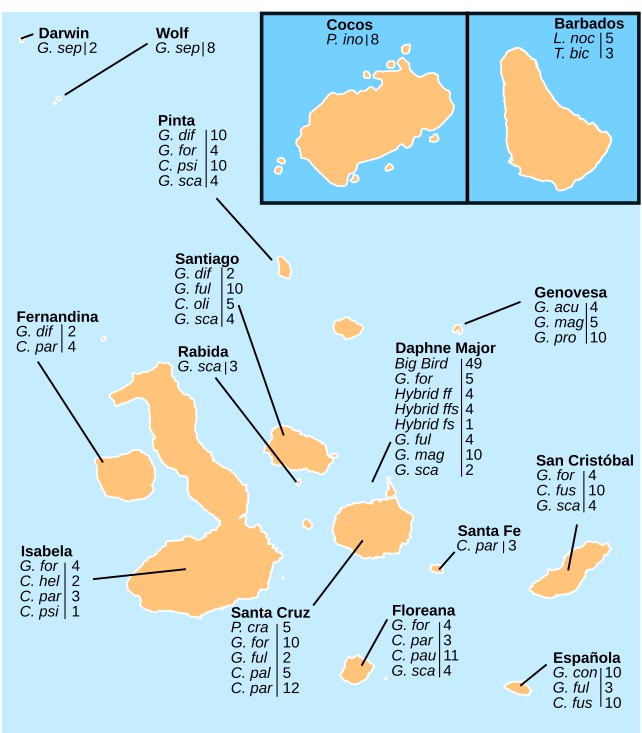

**Fig. 1 | Galápagos islands and sample locations.** A total of 293 Darwin's finch samples representing 18 species, 4 hybrids and 2 outgroup species (*Loxigilla noctis* and *Tiaris bicolor*) from 16 islands[6, 9–11] were included in the study. Island names are indicated above finch species, hybrids and their respective sampling sizes. Finch species abbreviations: **Big Bird** lineage (*Geospiza fortis* x *Geospiza conirostris*), **C. fus** (*Certhidea fusca*), **C. hel** (*Camarhynchus heliobates*), **C. pal** (*Camarhynchus pallidus*), **C. par** (*Camarhynchus parvulus*), **C. pau** (*Camarhynchus pauper*), **C. psi** (*Camarhynchus psittacula*), **C. oli** (*Certhidea olivacea*), **G. acu** (*Geospiza acutirostris*), **G. con** (*Geospiza conirostris*), **G. dif** (*Geospiza difficilis*), **G. for** (*Geospiza fortis*), **G. ful** (*Geospiza fuliginosa*), **G. mag** (*Geospiza magnirostris*), **G. pro** (*Geospiza propinqua*), **G. sca** (*Geospiza scandens*), **G. sep** (*Geospiza septentrionalis*), **Hybrid ff** (*Hybrid G. fortis* x *G. fuliginosa*), **Hybrid ffs** (*Hybrid G. fortis* x *G. fuliginosa* x *G. scandens*), **Hybrid fs** (*Hybrid G. fuliginosa* x *G. scandens*), **L. noc** (*Loxigilla noctis*), **T. bic** (*Tiaris bicolor*), **P. cra** (*Platyspiza crassirostris*), **P. ino** (*Pinaroloxias inornata*).

Understanding of how ERVs spread and distribute in host populations that have undergone speciation and retrovirus infection during the same time span is currently limited. Darwin's finches are well situated to provide a better understanding of this process, as they are a textbook example of an adaptive radiation and one of the best studied systems of concurrent hybrid and allopatric speciation[3]. There are 17 currently recognized species of Darwin's finches on Galápagos islands, and one species on Cocos island (Costa Rica)[4]. The ancestor of this monophyletic group split off from the ancestral group 1–2.3 MYA[4], during a period of climatic change with the closure of the Panamanian isthmus and the onset of the Pleistocene glaciation[5]. Adaptive radiation of the finches into a variety of ecological niches has resulted in the broad biological diversity observed today[6,7].

Recent genomic analyses have revealed ongoing ecological adaptation and gene flow between finch species[8]. This evolutionary history has been accompanied by retroviral infections, which have left their mark on the genomes of the present-day finch populations as ERVs. This genomic record provides an opportunity to investigate the utility of ERVs as a record of past retroviral infections to infer infection histories, and use ERVs as unique, age-directed sequence variants to trace gene flow between host populations post-speciation. Here, we take advantage of a recently released high-quality genome assembly of a Darwin's finch[9] to investigate the ERV abundance of 293 whole-genome sequenced individuals from across the Darwin's finch

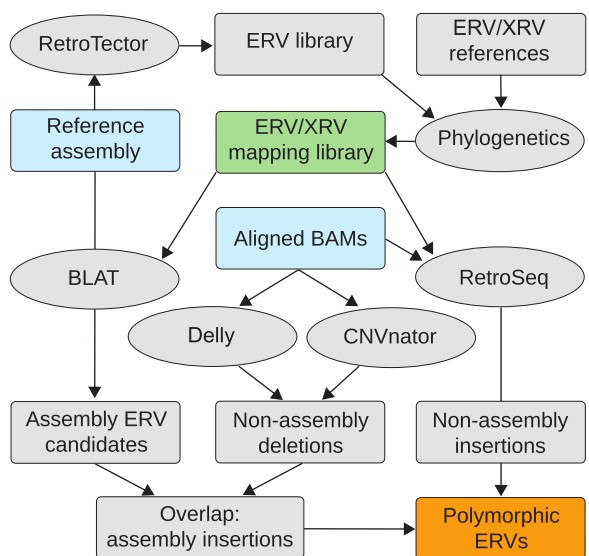

**Fig. 2 | Outline for identifying ERV segregation in host populations.** Overall strategy is to identify ERV anchored short reads that are located along host DNA. Flow-chart boxes indicate data and ovals indicate software. Briefly, an ERV mapping library (green box, Supplementary Data 1) is constructed using identified assembly ERVs from RetroTector[12], and placed in a phylogenetic context (Supplementary Data 2). Paired-end read mapping information in BAM files anchor ERV associated reads to reference assembly positions using RetroSeq[16] to identify non-assembly insertions. On the other end, DELLY[17] and CNVnator[18] are used to identify unique assembly insertions. All locations are then collected into an ERV loci data frame (polymorphic ERVs, orange box) to facilitate frequency estimates for segregating ERVs.

radiation and outgroup species, and reveal contrasting historical patterns of retrovirus-host dynamics.

## Results

### Sample information and reference assembly

Individual samples of Darwin's finches ($n = 285$) from Galápagos and Cocos islands, as well as individuals of two species (*Loxigilla noctis* and *Tiaris bicolor*) representing outgroups to the ancestor of Darwin's finches sampled from Barbados ($n = 8$), were previously whole-genome sequenced using short read, paired-end technology[6,10,11]. The samples represent all 18 species of Darwin's finches ($n = 226$) and four hybrid groups ($n = 59$). The finch species sampled on Galápagos were further subdivided by populations across 14 of the islands[6,10,11], as shown in Fig. 1. Alignments of the 293 whole-genome sequenced finch samples to the recently released assembly of the small tree finch (*Camarhynchus parvulus*, GenBank: GCA_902806625.1)[9] indicated 6–49x depth of coverage (median 15x; Supplementary Fig. 1).

### Identification of ERV loci in individual samples

To identify insertionally polymorphic (hereafter referred to as polymorphic) ERVs in the finch populations, as illustrated in Fig. 2, we first used the RetroTector software[12] to mine ERVs from the *C. parvulus* assembly. The 587 detected ERVs were then curated for proviral sequence completeness in a phylogenetic framework by comparing with known ERVs and exogenous retroviruses[13–15]. The resulting reference mapping library retained 132 finch assembly ERVs and 79 reference proviral sequences (Supplementary Data 1) representing the major retroviral clades (Supplementary Data 2), which we used in further analysis. Finch assembly ERVs were named cPa for *C. parvulus* ERV and individual loci were named in accordance with the proposed nomenclature in Gifford et al.[16] (Supplementary Table 1).

Joint analysis of the 293 sample alignments and the reference ERV mapping library above using RetroSeq[17], DELLY[18], and CNVnator[19]

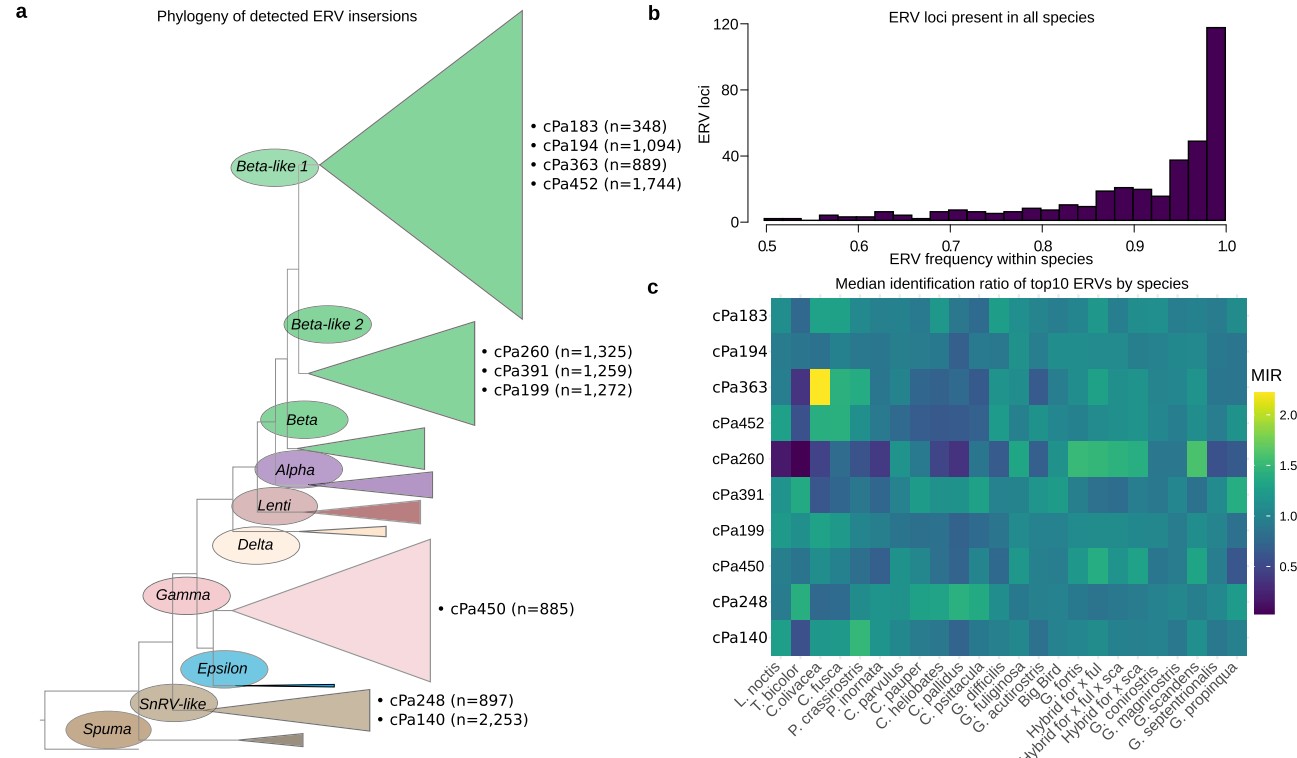

**Fig. 3 | ERV phylogeny and heatmap. a** Phylogeny of ERVs from the genome assembly together with retrovirus- and ERV-reference sequences establish evolutionary relationships and facilitate construction of a curated ERV mapping library to match unassembled short read sequences for ERV localizations along host DNA. Full phylogenetic tree is available in Supplementary Data 2. **b** Frequency histogram of ERVs at loci that contain at least one ERV identification in all species. Loci that fit this criterion are assumed to pre-date Darwin's finch speciation and are therefore expected to be fixed in all populations. Observed frequencies <1.0 of these ERVs, can be assumed to be the result of false-negative identification calls. **c** Heatmap showing varying ERV MIR in different, but closely related, host populations.

identified 26,964 putative ERV loci along the *C. parvulus* assembly coordinates. We observe considerable ERV polymorphism among the finches, which agrees with recent findings in mammalian host populations[15].

## Characterization of ERV loci

Associating ERV loci from the individual finches by similarity to sequences in the phylogenetic framework revealed a large expansion, compared to previous findings in e.g., chicken selection lines[20] and many other vertebrate assemblies[14], of *Beta-like 1* (n = 11,020), *Beta-like 2* (n = 5841), *Gamma* (n = 4264), and *SnRV-like* (n = 4326) ERV clades among Darwin's finches and their outgroup (Fig. 3a, Supplementary Data 2). These findings are consistent with previous pan-avian genomic estimates of increased *Beta-like* (LTR/ERVK) and *Gamma* (LTR/ERV1) counts in finches compared to other birds[21]. The number of loci associated with each cPa ERV ranged from 1–2253 (mean = 130, median = 35, s.d. = 283) and the number of ERV identifications within those loci among all samples ranged from 3–39,640 (mean = 2635, median = 823, s.d. = 5683). Linear regression analysis showed that the number of ERV loci correlates with number of identified ERVs among all samples ($r^2 = 0.81$, $p < 0.001$) despite the high variance in frequency of ERV identifications at each locus (mean = 0.003, $s = 0.056$) confirming a high degree of ERV polymorphism.

## ERV segregation patterns

ERVs that were present in all species accounted for only 1.2% of loci (334 of 26,964) and they were detected with an average frequency of 0.92 (Fig. 3b). We hypothesize that the oldest ERVs, which are detected in all species at the expected frequency of 1.0 for fixed loci, pre-date speciation of Darwin's finches, meaning that there is a considerable ERV variation dating post-speciation across the entire Darwin's finches'

radiation. The discrepancy between expected fixation (1.0) and observed mean frequency (0.92) indicates false negative rate at 0.08 of our detection method for homozygous ERV loci. While loci at high frequencies were called confidently, the identification sensitivity was reduced for low-frequency loci, which predominantly appear in the heterozygous state. Consequently, sequencing coverage is a strong predictor of ERV identification. Linear regression analysis showed correlation between ERV frequencies and sequencing read coverage in individual samples ($r^2 = 0.71$, $p < 0.001$), which indicated reduced ERV detection sensitivity in samples with lower sequencing coverage.

In order to determine the relative abundance of an ERV in an individual while controlling for sequencing coverage bias, we employed a standardization procedure. A ratio was calculated by dividing the number of identifications of each ERV type with the total number of identified ERVs in the analyzed individual, yielding a measure of relative abundance. When comparing ERV types between species, the median of an ERV's relative abundance within a species was divided by the median of that ERV's relative abundance in all finch samples. The resulting value, called the median identification ratio (MIR), indicates the relative between-species ERV abundance. MIR > 1 indicates that an ERV made up a greater proportion of ERV identifications in the focal species, compared to the proportion in the whole finch data-set, while MIR < 1 indicates the opposite. The relative ERV abundance, as expressed by MIR, reveals variation across finch species and ERV types (Fig. 3c).

The overall ERV distribution along chromosomes (Supplementary Fig. 2) indicate enrichment at either chromosomal end, albeit with a few exceptions for short chromosomes. Additionally, the ERV distribution between chromosomes is also notably non-random where a more than 30% difference in ERV density was observed between the larger chromosomes (Supplementary Table 2). These patterns suggest

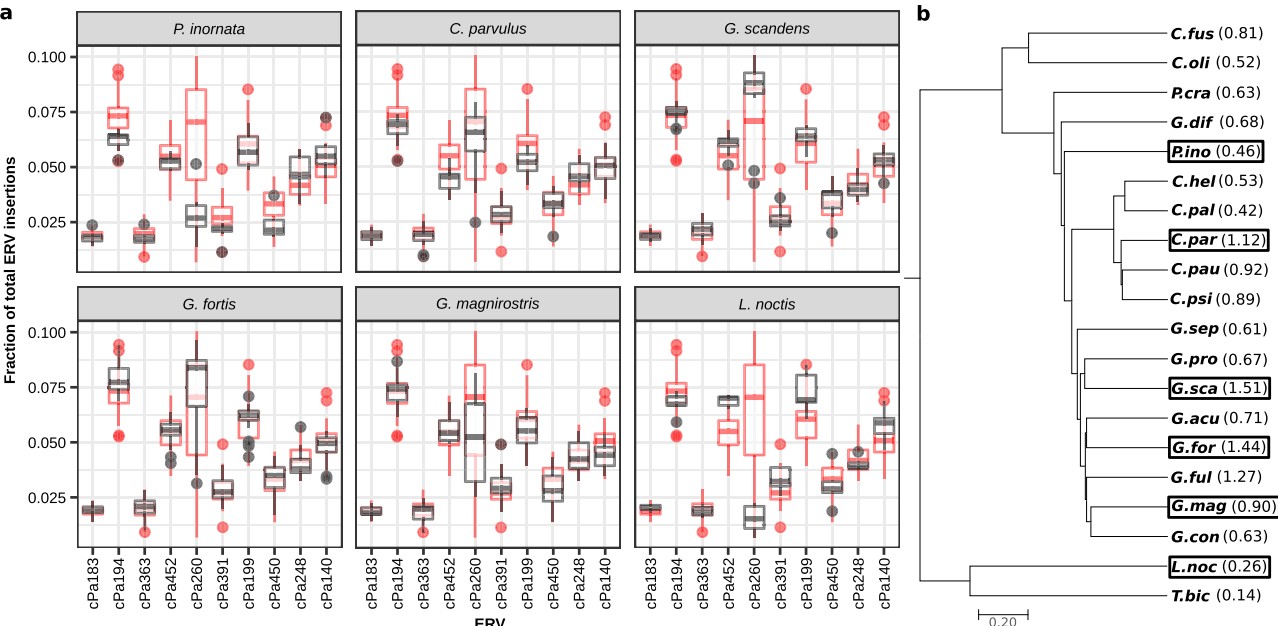

**Fig. 4 | Relative abundance of common ERVs in example species. a** The 10 most frequent ERVs showed significant variation in abundance both within and between species. The relative fraction of insertions of an ERV within an individual was plotted as a data point in the box plots. Red data points are for all finch samples, and black is the subset corresponding to only the species in the labeled window. The boxplot bottom and top hinges represent 25% and 75% confidence respectively, and whiskers indicate the 95% confidence interval. Variation in relative ERV abundance among all samples (e.g., cPa260), could be attributed to either within species variation (e.g., *G. fortis* and *G. magnirostris*), between species variation (e.g., *L. noctis* and *G. scandens*), or both. **b** Phylogeny of the non-hybrid Darwin's finch species with cPa260 MIR in parenthesis next to species abbreviations. Species in this phylogeny that also appear in panel **a** are highlighted with a border. Three of the four species enriched for cPa260 belong to the ground finch group, with the tree finch *C. parvulus* being the exception.

variable exposure or tolerance to ERV accumulation across the Darwin's finch genome.

## Abundant ERV association clusters

The 10 most frequent ERVs revealed significant variation both within and between species (Fig. 4a, Supplementary Figs. 3, 4). The *Beta-like 2* ERV, cPa260, showed the greatest variance of relative abundance between species. The cPa260 ERVs displayed significant species-effect on the proportion of cPa260 in individuals (one-way ANOVA test; $F(5,90) = 33$, $P < 0.001$), despite some species having considerable cPa260 variation between individuals, for instance *Geospiza fortis* (mean proportion cPa260 = 0.076, s.d. = 0.018). There were also notably fewer cPa260 ERV identifications in the outgroup species, *Loxigilla noctis* and *Tiaris bicolor*, relative to the Galápagos finch species. The individuals sampled from *L. noctis* had the smallest proportion of cPa260 (mean proportion cPa260 = 0.015, s.d. = 0.007). A change in retrovirus activity after Darwin's finch speciation in one or more of the resulting lineages could be responsible for producing this variation of cPa260 ERV insertions. The species carrying the largest proportional load of cPa260 were all species of ground finches, including *G. fortis* (mean = 0.076, s.d. = 0.018, MIR = 1.44), *G. fuliginosa* (mean = 0.072, s.d. = 0.016, MIR = 1.27), and *G. scandens* (mean = 0.085, s.d. = 0.014, MIR = 1.51) (Fig. 4b). The various hybrids of these species carried large proportions of cPa260 at the same loci as well, indicating that they inherited these ERVs from their parental species (Supplementary Fig. 4).

The *Beta-like 1* ERV, cPa452, showed notable variance where all tree finches (*C. heliobates, C. pallidus, C. parvulus, C. pauper, C. psittacula*) have relatively low fractions and both warbler finches (*C. olivaceae, C. fusca*) species have relatively high fractions. All ground finches (*G. acutirostris, G. conirostris, G. difficilis, G. fortis, G. fuliginosa, G. magnirostris, G. propinqua, G. scandens, G. septentrionalis*) including the hybrids have average fractions relative to background (Fig. 4, Supplementary Fig. 4). The low fraction of cPa452 in the tree finches is reflected evenly across the genome, whereas the higher abundance in the warbler finches appear driven by higher concentrations of insertions on chromosomes 1, 2, 5, 14, and Z (Supplementary Fig. 5).

The *Beta-like 1* ERV, cPa363, appeared more uniform than cPa260 across finch species with the exception of the green warbler finch (*C. olivacea*) where cPa363 were more than twice as frequent as the average for all finch species (Fig. 3c). The striking enrichment of cPa363 in *C. olivacea* was also significant when compared to all other species by Welch's two sample *t*-test ($t = 12$, df = 4.2, $p < 0.001$). At the loci where a cPa363 ERV was detected in *C. olivacea*, the ERV frequency in the population was considerable (total number of loci = 180, mean frequency = 0.51, s.d. = 0.30). On average, 4% of all ERVs identified in *C. olivacea* samples were cPa363. *C. olivacea*, and to some extent the related *C. fusca*, stands out regarding cPa363 MIR in pairwise comparisons between finches.

## ERVs in potentially adaptive regions

Though we assume a neutral mode of evolution for the majority of identified ERVs, potential effects on host genome function cannot be established from the currently available data. However, we investigated ERV frequencies in 28 genomic regions showing strong genetic differentiation between small, medium, and large ground finches, previously identified by Rubin et al.[9] We identified 379 ERV loci within these regions, 13 of which segregated at high frequencies in the large ground finch (*G. magnirostris*), and low frequencies in the small ground finch (*G. fuliginosa*), or vice versa (Table 1). The medium ground finch (*G. fortis*) was intermediate in frequency for these ERV loci. This segregation reflects the observed patterns in ground finches described by Rubin et al.[9], which indicates that low-frequency ERV variants were part of the selected haplotype.

Since Galápagos finch species are often subdivided into island populations (Fig. 1), we explored the ERV distribution across island populations. A modified MIR was used for comparisons between island finch species where the median ratio of an ERV's proportional

**Table 1 | ERV frequency within genomic regions associated with beak size**

| ERV locus | Genome position | ERV frequency at locus | | | |
|---|---|---|---|---|---|
| | | *G. magnirostris* | *G. fortis* | *G. fuliginosa* | *G. mag–G. ful* |
| ERV-GE.cPa583.28-CamPar | Chr2: 139,33,293 bp | 1,00 | 0,66 | 0,05 | 0,95 |
| ERV-S.cPa248.357-CamPar | Chr2: 126,790,181 bp | 0,93 | 0,69 | 0,00 | 0,93 |
| ERV-AB1.cPa529.231-CamPar | Chr2: 14,989,140 bp | 0,93 | 0,69 | 0,05 | 0,88 |
| ERV-AB1.cPa142.60-CamPar | Chr1A: 41,006,454 bp | 0,93 | 0,66 | 0,05 | 0,88 |
| ERV-AB1.cPa452.570-CamPar | Chr2: 15,047,613 bp | 0,87 | 0,41 | 0,00 | 0,87 |
| ERV-AB1.cPa452.1150-CamPar | Chr3: 107,769,927 bp | 1,00 | 0,53 | 0,16 | 0,84 |
| ERV-S.cPa248.238-CamPar | Chr1A: 33,361,744 bp | 0,80 | 0,50 | 0,00 | 0,80 |
| ERV-GE.cPa273.207-CamPar | Chr3: 60,476,691 bp | 0,87 | 0,59 | 0,11 | 0,76 |
| ERV-AB1.cPa255.18-CamPar | Chr2: 15,008,875 bp | 0,73 | 0,47 | 0,00 | 0,73 |
| ERV-AB1.cPa529.201-CamPar | Chr1A: 31,065,165 bp | 0,73 | 0,53 | 0,00 | 0,73 |
| ERV-AB1.cPa16.47-CamPar | Chr2: 127,077,908 bp | 0,00 | 0,13 | 0,42 | −0,42 |
| ERV-GE.cPa227.49-CamPar | Chr2: 22,491,398 bp | 0,00 | 0,09 | 0,58 | −0,57 |
| ERV-AB2.cPa199.745-CamPar | Chr3: 39,727,190 bp | 0,07 | 0,31 | 0,74 | −0,67 |

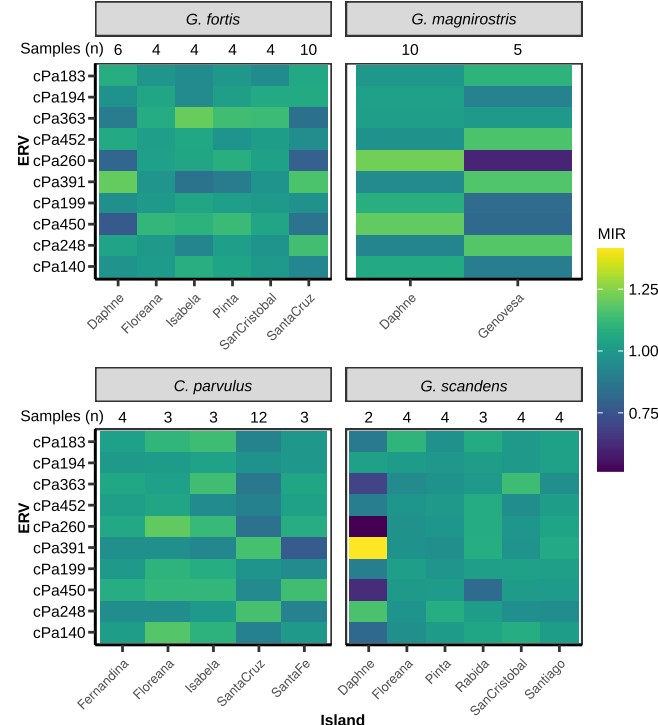

**Fig. 5 | Contrasting ERV landscapes across finches and islands.** Relative abundance of the 10 most frequent ERVs showed variation between island populations of the same species. Modified MIR normalized ERV abundance values both vertically across ERVs and horizontally across islands. A contrasting color between island populations for a given ERV indicates a difference in relative abundance between populations, either higher (lighter color), or lower (darker color). Variation in MIR increased with small sample size (e.g., *G. scandens* from Daphne island), however some contrasts are more likely to represent large actual differences in ERV abundance between island populations (e.g., cPa260 in *G. magnirostris*).

abundance in an island population of a species was divided by the median ratio of the ERV's proportional abundance for that entire species. Variation could be observed between islands (Fig. 5, Supplementary Fig. 5), but it remains unclear whether this was a consequence of retrovirus infection and colonization of germline as novel ERVs, or result from gene flow of ERVs and potential founder effects on the islands. The unexpected ERV variation in finch species on different

islands indicates historical changes in gene flow and selection between island populations in contrast to expected genetic drift towards loss or fixation of ERV loci within each species as a whole.

## Discussion

The role of infectious agents in the evolutionary history of a species can loom large, not just as a challenge to reproductive success, but also as a potential source of genetic innovation that allows for novel adaptations. Inferring that history is an enormous challenge, since the vast majority of those ancient agents are extinct, and surviving virus lineages have mutated and diverged extensively since their initial infection of the host species. The study of ERVs partially overcomes that challenge by utilizing the reduced rate of evolution within the host germline.

We observe many private and low frequency ERVs across the Darwin's finch populations, which agrees with recent estimations of polymorphic ERVs in wild and domestic animals[15,21,22]. About 1.2% of ERV loci are shared by all Darwin's finch species and are fixed, or nearly so, in the entire dataset. This is in contrast with loci at lower within-species frequencies, which are also shared among fewer species. Detection of high frequency, and mostly homozygous, ERVs was shown to be effective, with less than 8% estimated false negative rate. Conversely, detection of heterozygous ERVs was less powerful, resulting in a correlation between sequencing coverage and RetroSeq identification of ERV loci, which is an important consideration for invoking biological interpretations. Maximum sensitivity appeared to be reached at approximately 30x sequencing coverage and sensitivity decreased linearly with decreased coverage to about 55% false negative rate of heterozygous loci at 4x coverage. However, low frequency ERVs were still identified in high coverage species samples. The pattern appears accurate since these high confidence samples confirmed that most loci having low frequency ERVs also occur in single or small subsets of species. Segregating ERVs missing in some species account for 98.8% of all loci, which could be explained by incomplete lineage sorting at speciation and loss through drift rendering even old ERVs seemingly missing in some host species.

A conclusion from these analyses is that while *Beta-like-1*, *Beta-like-2*, and *Gamma* ERV clades saw large expansions in numbers throughout the Darwin's finch radiation, no single host species or monophyletic group of host species account for the overall increase in ERVs. With some notable exceptions discussed below, ERV enrichment does not follow an obvious pattern across the Darwin's finch radiation. This rules out the possibility that some extant descendants of the

original colonists of the Galápagos islands were especially susceptible to retroviral infection and ERV accumulation. Instead, our observations are consistent with the standing ERV variation being shaped by infection bursts distributed across the entire Darwin's finch radiation followed by demographic effects such as repeated bottlenecks and gene-flow. The most abundant of these ERVs allow us to infer some of this history due to a large number of integrations and relatively high segregating frequency in some species. The variation between the warbler-, tree- and ground finch groups in, for example, cPa452 ERV distribution along the genome indicates temporal variations in host tolerance to ERV accumulation or genetic drift during the Darwin's finch radiation. Individual variation of ERVs within populations showed intriguing patterns that suggested recent, possibly ongoing, retroviral activity. When compared to other abundant ERVs, the cPa260 ERVs stand out, showing a high degree of variation both between species and within species. *Geospiza magnirostris* in particular has high within-species variation, where the cPa260 proportion in individuals ranged from 2.5% to 8% of total ERVs. We also note that the *G. magnirostris* sampled on Daphne were enriched for cPa260 ERVs compared to the Genovesa population. The outgroup species show lower cPa260 counts compared to the finch species on Galápagos, yet even there it exists at low frequency at all loci. This pattern suggests recent retroviral infection after divergence between Darwin's finches and the outgroups, 1–2.3 MYA[4]. The cPa363 ERV also displayed a striking enrichment in a single species, the green warbler finch, *C. olivacea*. Our *C. olivacea* samples (*n* = 5) came from the island of Santiago, where we also had samples from *G. difficilis*, *G. fuliginosa* and *G. scandens*, however, these did not show cPa363 enrichment. This may indicate a greater activity by the corresponding retroviruses in *C. olivacea* contributing to the observed cPa363 ERV accumulation. However, the observed pattern across species should be viewed with some caution; although each individual MIR calculation is supported by observations across loci, sample size in terms of individuals per species remains limited. The significance of these observations is currently unclear, but as *C. olivacea* is also present on a number of other Galápagos islands together with other finch species, further sampling across these species and islands, as well as increased sequencing coverage could improve our insight into the cPa363 invasion of the *C. olivacea* genome. Many ERV groups displayed minor MIR variations across the species. These observations suggest a shared history of retrovirus-host activity both between species and within populations, which is plausible considering the proximity of Galápagos islands and well-documented inter-island migration and admixture[6,23].

Future studies can take advantage of ERVs to increase the resolution of, for instance, the Darwin's finch phylogeny by leveraging the directional nature of ERV evolution within the host genome. Unlike point mutations that can revert between states and be subject to multiple changes over time, structural ERV variation can exist in three forms proceeding from: (1) empty pre-integration site, to (2) proviral integration, to (3) solitary LTR following homologous recombination between the two proviral LTRs. Thus, a shared ERV locus between hosts provides strong evidence for shared ancestry of a genomic region, as shown by for instance cPa452 variance across warbler-, tree- and ground finches (above). Neutral ERV allele frequency can be applied as a proxy for ERV locus age as a function of population size and mutation rate; however higher sequencing coverage would be required for a confident estimate.

The potential role of selection in shaping ERV variation is currently under investigation. Allele frequencies at the 13 ERV loci described in Table 1 covary with body and beak sizes among the ground finches[9]. While a possible causal role of an ERV locus in beak and/or body size phenotypes remains to be explored, we conclude that at least part of the identified ERV variation described herein has been shaped by selection or more likely through hitchhiking on linked variants. The outcome is ERVs with large frequency differences between closely related ground finch species at critical genomic locations.

In summary, we identified considerable ERV variation across populations of Darwin's finches, uncovering contrasting ERV landscapes that reflect historical differences in retrovirus-host dynamics, as well as indications of historical changes in gene flow and selection based on the unexpected variation in finch species on different islands. The Darwin's finch retrovirus-host system, with host speciation in progress, represents a natural model for evaluating the extent and timing of retroviral activity in hosts undergoing phylogenetic radiation and colonization of new environments, as well as future studies of the extent to which ERVs contribute to host biology by connecting novel and segregating ERVs with adjacent host genes and phenotypes across host populations.

## Methods

### Finch data

Individual samples of Darwin's finches (*n* = 285), distributed among 18 species (*n* = 226) and four hybrid groups (*n* = 59), as well as individuals of two species from Barbados (*n* = 8) representing outgroups to the ancestor of Darwin's finches, were previously collected from Galápagos and Cocos islands of the Pacific and Barbados in the Carribean[6,11]. The finch species sampled on Galápagos were further subdivided by populations among 14 of the islands, although all hybrid populations were only sampled from Daphne Major (Fig. 1).

Illumina paired-end reads[6,10,11,24] were mapped to the small tree finch reference assembly (Genbank: GCA_902806625.1) using BWA-MEM[25], indexed and sorted using SAMtools[25], and Picard (http://broadinstitute.github.io/picard) was used to mark duplicates within alignments prior to further analysis.

### ERV mapping library

The RetroTector v1.0 software[12] was used to mine ERVs from the small tree finch assembly (Genbank: GCA_902806625.1). We identified 587 ERVs that were curated for completeness with additional proviral reference sequences in a phylogenetic framework using FastTree2 v2.1.7[26], applying the GTR model of nucleotide sequence evolution, based on sequence alignment of amino acid motifs in the *gag* (two motifs in matrix; two in capsid; two in nucleocapsid), *pro* (two from protease), and *pol* (11 in RT and 9 in IN) genes[13–15], in order to construct an ERV mapping library. We then pruned the library for ERVs with both 5′- and 3′-LTRs that did not bridge assembly scaffold boundaries. The curated mapping library contained 132 ERVs, as well as 79 reference proviral sequences (Fig. 2, Supplementary Data 1) spanning the retroviral phylogeny (Fig. 3a, Supplementary Data 2).

### Non-assembly ERV insertion mapping

RetroSeq[16] (https://github.com/tk2/RetroSeq; accessed 2020-02-18) discover was applied to each mapped BAM file with the align option to the curated ERV mapping library (Supplementary Data 1). RetroSeq call was applied on the intermediate output with the soft-clip option and at least 5 reads per call. Calls with stringent RetroSeq filters (FL 8; min GQ 10; max GQ 200; clip3 2; clip5 2) were used to define putative ERV loci in R 3.6.3[27]. Insertion locations were extended by 50 bp in both directions, and reduced in GenomicRanges 1.38[28] to a list of 26,266 putative ERV loci. RetroSeq call was then rerun with the soft-clip option and only 1 read requirement with relaxed filters (FL 5) when counting ERVs in each finch sample. ERV presence, mapped location, and annotation by RetroSeq were compiled for each locus and ERV frequency was calculated within each finch species. The coordinates of each locus identified by RetroSeq correspond to the empty pre-integration site, which is lacking an ERV in the reference genome assembly.

## Assembly ERV insertion mapping

CNVnator 0.3.3[19] and DELLY 0.7.7[18] were used to identify assembly-specific ERVs that showed polymorphism (i.e., assembly-specific ERVs that were not present in the sequenced finch individuals, identified as a "deletion" in an individual relative to the reference individual). These polymorphisms were identified by finding overlaps between deletions calls by CNVnator and DELLY and blocks of similarity to the ERV mapping library as identified by BLAT v. 36[29].

CNVnator was applied with 300 bp bin sizes to call structural variants in each individual. Deletion calls were filtered (eval1 ≤ 0.05; eval2 ≤ 0. 05; max length 20,000 bp), overlapping calls were reduced using GenomicRanges in R, and subset to those that overlapped regions of the genome with ERV similarity, as determined by BLAT, resulting in 207 putative ERV loci. Filters were then relaxed (eval1 ≤ 1 and eval2 ≤ 1) in order to count ERVs across the finch samples.

Samples were individually called by DELLY and merged into a single vcf file using BCFtools merge[25]. The vcf was read into R using intansv[30] and VariantAnnotation[31] packages. Regions were filtered for length (90–20,000 bp) and at least two supporting reads. Overlaps between DELLY loci and regions of the genome showing ERV similarity, as determined by BLAT, were identified using GenomicRanges in R, resulting in 491 putative ERV loci. DELLY genotypes were used to count ERVs across the finch samples.

## ERV segregation in host populations

A total of 26,962 ERV loci (26,266 non-assembly RetroSeq calls and 698 assembly-specific CNVnator and DELLY calls) were included in a data frame for analyses of ERV segregation across the finch populations (Fig. 1). Custom R 4.1.1[27] scripts were used to calculate ERV-insertion frequencies and distribution among species and islands. Plots were generated using Tidyverse[32].

## Statistical methods

ANOVA F values are reported in results section. Welch's two sample *t*-test was applied as a two-tailed test to determine the significance of cPa363 population mean difference in *C. olivacea* from other species. Whisker length in Fig. 4 and Supplementary Fig. 3 correspond to approximately 95% confidence intervals, however, direct comparisons of ERV abundance were tested using ANOVA and Welch's two sample *t*-test as described above.

## Reporting summary

Further information on research design is available in the Nature Research Reporting Summary linked to this article.

# Data availability

The small tree finch assembly is available at Genbank: GCA_902806625.1[9]. Illumina sequencing data is available from the Sequence Read Archive (www.ncbi.nlm.nih.gov/sra) under BioProject PRJNA743742.

# Code availability

Code and supporting files are available at GitHub: (https://github.com/PatricJernLab/Darwins_finches_ERV_diversity; https://doi.org/10.5281/zenodo.7116320).

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

## Acknowledgements

We thank Sangeet Lamichhaney, Fan Han, and Erik Enbody for discussions and assistance in accessing finch sequences and samples. This work was funded by Swedish Research Council VR (Grants 2018-03017; 2021-01740) and FORMAS (Grant 2018-01008) to P.J. The computations and data handling were enabled by resources provided by the Swedish National Infrastructure for Computing (SNIC) at UPPMAX partially funded by the Swedish Research Council through grant agreement no. 2018-05973.

## Author contributions

P.J. and L.A. conceived the study. M.L., J.H., M.E.P., and P.J. designed the study and analyzed data. ML and JH developed and performed bioinformatic analyses with input from M.E.P. and P.J. C.J.R. generated the genome assembly and contributed to interpretations. B.R.G., P.R.G., and L.A. contributed data and interpretations. J.H., M.L., and P.J. wrote the manuscript with comments from the other authors. All authors approved the manuscript before submission.

## Funding

## Competing interests

The authors declare no competing interests.
