## [Peer Review File · Nature Communications]

Spatiotemporal variations in retrovirus-host interactions among Darwin's finchesReviewers' Comments:

Reviewer #1:

Remarks to the Author:

Hill et al. report an investigation of endogenous retroviruses (ERVs) in Darwin's finches. The authors performed a large-scale screening of ERVs in 18 species of Darwin's finches and in two closely related outgroup bird species, *Loxigilla noctis*, and *Tiaris bicolor*. The authors have identified many potential polymorphic ERVs, and relative abundances of various ERVs are reported, analysed, and discussed, while accounting for sequencing coverage bias and potential missing data. However, ERV classification and identification in this work may need some more careful consideration and attention. Also, given the title of the work "Spatiotemporal variations in retrovirus-host interactions among Darwin's finches", some additional analyses should be performed and the results should be further discussed.

Comments

ERV naming and grouping

1) Please renaming all of the identified ERVs using the standard ERV nomenclature. See Gifford et al., Nomenclature for endogenous retrovirus (ERV) loci, 2018, *Retrovirology*.

2) Line 90: the authors report that they have identified 26,962 putative ERV loci. However, a quick look at the supplementary table appears to suggest that there were 26,964 putative ERV loci found. Please check.

3) Lines 99–100: Related to the comment above, the authors report that "[t]he number of loci associated with each cPa ERV ranged from 4 – 2,252 (mean = 189, median = 65, s.d. = 339)..." However, the way the results are compiled in the supplementary table makes it difficult to cross check these numbers. For example, by counting the ERV assignments in the column `erv.assigned`, at face value, it appears that there are many ERV groups with a single member existing at just 1 locus, lower than 4 as reported by the authors. On the other end, I found that ERVs uniquely similar to cPa140 is the most expanded group of ERVs; however, they appear to exist at just 2,198 uniquely identifiable genomic loci, and not 2,252 loci as reported. If I were to count all ERVs that show similarity to cPa140, although not uniquely, I would get 2,255, and again not 2,252. Please check.

4) Based on the supplementary table, it appears that an ERV could be identified as similar to multiple reference ERVs. An ERV at the locus `RETROSEQ_chr1:4911537-4911637`, for example, is assigned to be in the group "cPa140|cPa260", which, if I understand correctly, means that it is considered similar to both cPa140 and cPa260 (right?). Although the authors have stated that ERV assignment was based on sequence similarity; however, I feel that this information is not sufficient for me to fully understand or reproduce the analysis. Specific details about group demarcation and similarity cut-off, for example, were missing. Please provide more details regarding how the ERV assignment was done, and clarify the existence of ERVs assigned to multiple reference ERVs.

5) Related to this matter, it is unclear how the authors treated ERVs assigned to multiple reference ERVs in their relative abundance calculation. Would an ERV in the group "cPa140|cPa260", for example, counted as a cPa140 ERV, or a cPa260 ERV, or both? If both, were the counts weighted in any way? etc. Please clarify.

6) In addition, the fact that such ERVs exist leads me to suspect that reference ERVs, especially those newly identified by the authors in the small tree finch, likely have a strong underlying phylogenetic structure (as indeed shown in Fig 3) and could be better classified. Given the systemic nature of the work, I strongly recommend the authors to perform a detailed systematics analysis and better classify all of the ERVs analysed, starting from reference ERVs. This might allow them to uniquely assign each ERV to a non-overlapping ERV group, benefiting downstream analyses and improving result interpretation.

7) Interestingly, based on Fig 3, it is clear that cPa140 and cPa260 are phylogenetically very distinct, belonging to very different retroviral genera, and yet, an ERV could be assigned to be similar to both of them at the same time... Examination of the supplementary table in more detail revealed many of these similar occurrences. I also found that >92% of the putative ERV-containing loci identified are less than 2 illumina-read long, being 300 bp in length or shorter (24,811 loci / 26,964 loci), and only <2% of the loci were $\geq 1,000$ bp long (459 loci / 26,964 loci). I'm not sure how many of these are empty integration sites in the reference genome or are just very short sequences identified as ERVs. Specifically, I'm unsure how an ERV that is not present in the reference genome could be confidently classified and identified based on a reference-based read mapping analysis (Lines 264-272). This raises a serious concern that many of the ERVs identified could be false positive, and may in part explain the observed ambiguous ERV group assignments. Please clarify and explain the ERV characterisation protocol in more details to justify that they are not false positive and could be classified / identified confidently.

ERV integration site distribution

8) Lines 129-134: These findings are rather interesting, but the authors hardly discuss the results. I would like to see the authors compare and contrast these findings against those of other ERVs, like human ERVs, and perhaps further discuss the nature of ERV integration / selection process.

ERV and host co-evolution

9) The authors report the host distributions of some ERVs in detail, e.g. those of cPa260 ERVs (Lines 135-149), cPa452 ERVs (Lines 150-157), and cPa363 ERVs (Lines 158-166). However, the discussions could be much richer, and evolutionarily more meaningful if they were done in the host evolutionary context, which is missing entirely from the work. One suggestion is to estimate the host phylogeny, and map the ERV abundances onto the host tree, or track the ERVs' radiation along the host speciation process to see how they co-diversify along each other, for example.

10) Despite the name of the work "Spatiotemporal variations in retrovirus-host interactions among Darwin's finches", spatio- (Lines 167-177) and temporal- (Lines 202-224) variations of the ERVs are minimally investigated. In fact, direct temporal information of the ERVs is missing entirely from the work, and their potential temporal variations are only minimally discussed, inferred indirectly from varying ERV's host distributions. Direct comparison of the ERV and host age estimates in the context of the virus and host co-phylogenies and spatial distributions could be highly informative and helpful with the discussion of the virus-host co-evolutionary process. Although the authors mention that "higher sequencing coverage would be required for a confident [age] estimate" (Lines 233-234); this should be possible for their top 10 ERVs at least.

Minor comments

1) Lines 264 and 273: I found the phrases "Non-assembly ERV insertion mapping" and "Assembly ERV insertion mapping" rather confusing. Please consider changing them to something else more reflective of the nature of the analysis aims.

2) Fig 4 and Fig S3: Please sort their x axis to be in the same order.

3) The term 'polymorphic' in this work appears to specifically mean 'insertionally polymorphic', and doesn't refer to any other kinds of polymorphisms, like nucleotide or structural polymorphisms. If so, please change the term throughout the text to be more specific and precise.

Reviewer #2:

Remarks to the Author:

In this manuscript, Hill et al. explored the diversity and evolution of endogenous retroviruses (ERVs) in Darwin's finches that represent a text-book example of adaptive radiation. Briefly, the study

investigated the diversity of ERVs in Darwin's finches, and reported the insertional polymorphism of ERVs across these finches. However, I don't see many interesting results expected when reading the title (for example, the relationship between the evolution of ERVs and the adaptive radiation of Darwin's finches). In my opinion, the analyses seems to be more or less superficial and might not represent the advances required for a Nature Communications paper.

Major comments:

1. Line 82-88. It remains unclear why a reference ERV mapping library should be reconstructed. Based on the context, I guess a reference ERV represents a virus that invaded the finch genomes.
2. Line 89-92: The logic for mining ERV polymorphism is not well explained. Its performance appears to be obscure. I suggest an alternative pipeline that mined ERVs in each individual first and assigned orthologous relationships, which is much more straight-forward. Of course, there are other ways. Moreover, I suggest a figure showing ERV polymorphism across individuals could be provided.
3. The ERV polymorphism is not well connected with host population history. I guess the authors tried, but failed to figure out. Maybe simulation work will provide some insights.

Minor comments:

L80: the recently released assembly of the small tree finch should be introduced, because it is a highly contiguous assembly.

L95-96: These numbers are not equal to 26,962.

L99-101: The meaning of these numbers is unclear. Please clarify.

Reviewer #3:

Remarks to the Author:

NCOMMS-22-09493

This paper investigates the diversity and distribution of endogenous retroviruses (ERVs) in the genomes of Darwin's finches and close relatives. They leverage a large dataset of sequenced genomes from more than 20 species to characterize ERVs and document patterns of diversity and relics of host-viral interactions. The authors report a large diversity of ERVs among these species and highlight a number of interesting patterns. Few ERVs (< 2%) were found in all species, suggesting that the spread of ERVs is dynamic and much diversification has happened since the radiation of finches. They also draw attention to technical sources of variation, namely that there is a strong relationship between sequencing coverage and ERV diversity, and that the sample size of individuals sequenced from a given population also appears to affect ERV diversity. I think this is an interesting topic and there is a lot of promise in this dataset.

The authors promote the strength of the Darwin's finch system for studies of evolution. However, I think the manuscript is lacking integration of the ERV results with the biological context. I'm not clear about exactly why Darwin's finches are an important and revealing system for this study, or exactly what new conceptual ground these data cover. I didn't have a clear understanding from the introduction how the authors were planning to use their investigation of ERVs to solve outstanding questions about either finch evolution and/or host-pathogen coevolution. I understand that the authors may have prepared their manuscript with word limits in mind; however, I think that there are several areas where either expanded background or discussion would assist the reader's understanding of the importance of this study.

For example: The authors suggest that this screening of ERVs in the Darwin's finch radiation helps "evaluate aspects of the natural history of the entire Darwin's finches' radiation." I felt any mention of natural history in the discussion was lacking. Can the authors elaborate?

The authors investigate ERVs in both “pure” species as well as several hybrid lineages. Why were hybrids included in this study? What do ERVs reveal in those lines? On line 147 the authors note that hybrids of *Geospiza* finches have cPa260 at the same loci as the parental species, indicating that these ERVs are transmitted vertically. I would assume though that ERVs are heritable so I’m left wondering what is revealing about sequencing these hybrids?

Line 255: The authors provide some tantalizing ideas of how ERVs could be used to further resolve the Darwin’s finch phylogeny. I understand that was not part of the current study, but I think that synthesis along just those lines would help integrate these new data within a broader context about ecology and/or evolution.

ERVs are neat in studies of evolutionary biology because unlike point mutations they are unlikely to revert. Thus, the authors point out that two species that have a common ERV almost certainly inherited it from a common ancestor (i.e. convergence of ERVs is unlikely). However, with so many of the ERVs private to certain species I was left wondering how common loss of ERVs is. Can the authors provide any more information about how much to expect the loss of these variants to create the observed patterns and whether loss can ever be definitively inferred? For instance, the authors report that one of the outgroup birds, *L. noctis*, had a small proportion of cPa260, an ERV that is more common in Darwin’s finches. The authors suggest that changes in viral activity post speciation could have produced the variation of cPa260 in the Darwin’s finch radiation. Could it also be possible that selection or population bottlenecks in *L. noctis* could have reduced variation in that species?

How much of these patterns could be explained by demographic history? For instance, does population size affect ERV diversity? e.g. *G. fortis* on Santa Cruz or Isabela has a very large population, but would be substantially smaller on Pinta. Same thing *C. parvulus* on Santa Cruz vs. Santa Fe). Is ERV diversity correlated with population size or even something as crude as island size? How much of the ERV diversity should we expect to be a product of antagonistic co-evolution between hosts and pathogens vs. a product of neutral processes of drift etc?

I think that integration of these results with the evolutionary history of this radiation is a bit lacking. For instance, we know from recent genomic work (Lamichhaney et al. 2015/16) that warbler finches diverged first in the Darwin’s finch radiation and the Cocos finch emerged later and ultimately ended up outside the Galapagos on Cocos Island. Could these evolutionary relationships be used to infer something about the transmission patterns of ERVs?

Minor comments:

Line 158: *C. olivacea* is highlighted here as an outlier finch that was enriched for cPa363. This species also seems to be lightly sampled (N = 5) compared to the rest of the species. Do the authors have any reason to believe that sampling artifacts could have produced this result?

Line 209: Differences in cPa260 ERVs in the *G. magnirostris* populations on Genovesa and Daphne populations: There was a large difference in sample sizes of these two populations (10 vs. 5). Could that explain some of the difference?

Fig 3b. Apologies, I’m not an expert in this field but I don’t understand how to interpret this panel. Could the authors consider rephrasing or expanding the legend?

Fig S3: Recommend adding genus abbreviations before species name (e.g. “*G. fortis*”)

Reviewer #4:
Remarks to the Author:

The manuscript submitted by Hill et al describes a unique study that leverages finch genome data and recent finch genome assembly and the rich knowledge base centering on the natural history of Darwin's Finches in the Galápagos islands to explore the population genetics of endogenous retrovirus loci (ERVs) overlapping in time with the radiation of new species/lineages. While on the surface the study is mostly a descriptive cataloguing, it potentially provides a wealth of insight into virus-host interactions as they play out over generational timescales. That is to say, this manuscript could lay important groundwork for follow up studies of these data and for similar situations in nature where species history and corresponding sequence data for multiple closely allied species are available. One suspects that treating the individual ERV insertions as markers might ultimately provide insight into difficult-to-resolve host relationships. While much of that is beyond this first study, this paper lays the groundwork.

I have a couple of minor comments to aid in accessibility of the manuscript to the variety of readers who might be interested:

Some brief explanation of the "beta-like", "gamma-like", etc., terminology would be useful - even just a sentence with appropriate citations.

I encourage the authors to consider being clearer in distinguishing between "ERV" as a group, "ERV" as a specific insertion/locus.. For example, in the Discussion in lines 190 and 192, it would be more precise and less confusing to say "...homozygous, ERV loci..." rather than homozygous ERV, and similarly "heterozygous ERV loci" in line 192. This can also be an issue when discussing variation, frequency, abundance, etc., where it needs to be clear whether one is speaking of variation at a specific locus across a population versus variation within a group of ERV comprising many loci distributed across a genome.

Line 199-201 - does this imply that ongoing or recent infection by retroviruses closely overlapped in time with the arrival and early radiation of the finches? Seems like this might be worth more discussion. Also, might the authors include some speculation as to the possible impact of the viruses on evolution of the host lineages? It would seem that these data are well suited for asking questions about the potential contributions of ERV to genome evolution.

Dear Reviewers,

We thank you for the encouraging and constructive comments to improve our manuscript. We have responded to the requests and comments point-by-point below, and updated the text, figures, and the supplementary data accordingly.

Patric Jern

Reviewer #1 (Remarks to the Author):

Hill et al. report an investigation of endogenous retroviruses (ERVs) in Darwin's finches. The authors performed a large-scale screening of ERVs in 18 species of Darwin's finches and in two closely related outgroup bird species, *Loxigilla noctis*, and *Tiaris bicolor*. The authors have identified many potential polymorphic ERVs, and relative abundances of various ERVs are reported, analysed, and discussed, while accounting for sequencing coverage bias and potential missing data. However, ERV classification and identification in this work may need some more careful consideration and attention. Also, given the title of the work "Spatiotemporal variations in retrovirus-host interactions among Darwin's finches", some additional analyses should be performed and the results should be further discussed.

Comments

ERV naming and grouping

1) Please renaming all of the identified ERVs using the standard ERV nomenclature. See Gifford et al., Nomenclature for endogenous retrovirus (ERV) loci, 2018, *Retrovirology*.

We consulted with two senior authors on the proposed nomenclature referred to here, to find a suitable adaptation for our current study. Note that the proposed nomenclature is considered optional by the authors since it is neither a ratified standard nor widely adopted. The intended purpose was to enable "accession reference" to individual loci when comparing primarily genome assemblies, which is ideal for small numbers of well-characterized loci but less so for studying segregating loci in large-scale population screening as in our current study. However, we acknowledge the advantage of introducing individual loci identifiers, to facilitate future studies. We therefore include a nomenclature conversion in the new Supplementary Table 1 and updated our ERV data frame accordingly with unique loci identifiers modified from the proposed nomenclature.

Text changed starting at line 90:

The resulting reference mapping library retained 132 finch assembly ERVs and 79 reference proviral sequences (Supplementary Data 1) representing the major retroviral clades (Supplementary Data 2), which we used in further analysis. Finch assembly ERVs were named cPa for *C. parvulus* ERV and individual loci named in accordance with the proposed nomenclature in Gifford *et al.*¹⁶ (Supplementary Table 1).

2) Line 90: the authors report that they have identified 26,962 putative ERV loci. However, a quick look at the supplementary table appears to suggest that there were 26,964 putative ERV loci found. Please check.

We confirmed 26,964 loci as being the correct count, and updated the text accordingly. Two DELLY loci had mistakenly not been included in the reported totals but were present in the analyses.

3) Lines 99–100: Related to the comment above, the authors report that "[t]he number of loci associated with each cPa ERV ranged from 4 – 2,252 (mean = 189, median = 65, s.d. = 339)..." However, the way the results are compiled in the supplementary table makes it difficult to cross check these numbers. For example, by counting the ERV assignments in the column `erv.assigned`, at face value, it appears that there are many ERV groups with a single member existing at just 1 locus, lower

than 4 as reported by the authors. On the other end, I found that ERVs uniquely similar to cPa140 is the most expanded group of ERVs; however, they appear to exist at just 2,198 uniquely identifiable genomic loci, and not 2,252 loci as reported. If I were to count all ERVs that show similarity to cPa140, although not uniquely, I would get 2,255, and again not 2,252. Please check.

The reviewer correctly ascertained that there were ERVs with fewer than 4 loci, which were not included in the statistics. When reporting the range of insertion loci per ERV we counted only the 'cPa' ERVs which were (1) found in the reference assembly and (2) not identifiable as one of the reference sequences used to establish the ERV phylogeny. This was not stated clearly in the text, however upon further consideration those loci which were identified as similar to reference ERVs should be included in the statistics and the text has been updated accordingly.

When designating the identity of an ERV locus we used the majority consensus of the reads at that locus containing ERV sequence as determined by the particular method (RetroSeq, CNVnator, or DELLY). The 'erv.assigned' column in the supplementary data contained some cases where secondary identifications were included which were later removed in the analysis script. The ERV identification with the highest support is listed first, and when only this ERV is considered the number of loci reflects what is listed in the text. However, upon closer examination, the maximum number of loci should be stated as 2,253, not 2,252 and the text has been updated to reflect that. The inclusion of non cPa ERVs in this section of the analysis impacted the immediately following statistics as well.

Lines 101-107 updated as follows:

The number of loci associated with each cPa ERV ranged from 1 – 2,253 (mean = 130, median = 35, s.d. = 283) and the number of ERV identifications within those loci among all samples ranged from 3 – 39,640 (mean = 2,635, median = 823, s.d. = 5,683). Linear regression analysis showed that the number of ERV loci correlates with number of identified ERVs among all samples ($r^2 = 0.81$, $p < 0.001$) despite the high variance in frequency of ERV identifications at each locus (mean = 0.003, $s = 0.056$) confirming a high degree of ERV polymorphism.

4) Based on the supplementary table, it appears that an ERV could be identified as similar to multiple reference ERVs. An ERV at the locus RETROSEQ_chr1:4911537-4911637, for example, is assigned to be in the group "cPa140|cPa260", which, if I understand correctly, means that it is considered similar to both cPa140 and cPa260 (right?). Although the authors have stated that ERV assignment was based on sequence similarity; however, I feel that this information is not sufficient for me to fully understand or reproduce the analysis. Specific details about group demarcation and similarity cut-off, for example, were missing. Please provide more details regarding how the ERV assignment was done, and clarify the existence of ERVs assigned to multiple reference ERVs.

Identification of ERVs with RetroSeq utilizes information from soft clipped ends of mapped reads that indicate an ERV integration and the unmapped mates of short reads which mapped adjacent to an ERV integration. ERV integrations were confirmed via sequence similarity of these soft-clipped sequences and unmapped reads to our ERV reference library, which we generated from the genome assembly and an identification assigned based on sequence similarity. Based on the short-read sequencing and the sequencing library size, it's most likely that our candidate ERV integrations mainly rely on sequence similarities within the LTR region. The noncoding LTRs can share sequence similarities with distantly related ERVs by chance, and thus we may find ERV loci that are assigned to distantly related groups, as the reviewer

pointed out. Since our aim in this study is to observe the bulk segregation patterns of ERV's in a natural population, and the vast majority of ERV loci were identified as similar to a single reference sequence with high confidence, we believe that a small number of potential misidentifications do not impact the analysis and results in a way that would alter the conclusions.

5) Related to this matter, it is unclear how the authors treated ERVs assigned to multiple reference ERVs in their relative abundance calculation. Would an ERV in the group "cPa140|cPa260", for example, counted as a cPa140 ERV, or a cPa260 ERV, or both? If both, were the counts weighted in any way? etc. Please clarify.

In the case where the sequence of short reads that were attributed to a particular locus had similarity to conflicting reference ERVs, the reference with the plurality of most similar reads was attributed to the locus. Other minority ERV assignments were discarded and only the primary assignment was used in downstream analysis.

6) In addition, the fact that such ERVs exist leads me to suspect that reference ERVs, especially those newly identified by the authors in the small tree finch, likely have a strong underlying phylogenetic structure (as indeed shown in Fig 3) and could be better classified. Given the systemic nature of the work, I strongly recommend the authors to perform a detailed systematics analysis and better classify all of the ERVs analysed, starting from reference ERVs. This might allow them to uniquely assign each ERV to a non-overlapping ERV group, benefiting downstream analyses and improving result interpretation.

Here we investigated differences in ERV loci across the finch radiation, across many individuals, islands and species. A detailed virus systematics and nomenclature investigation as suggested is beyond the scope of our study, and would not change our insights significantly. We updated the ERV loci dataframe with unique identifiers (see above) based on the present identifications to facilitate future studies. Mapping variation is not a function of the ERV phylogeny, which is derived from the *gag*, *pro*, and *pol* genes, but depends on coverage into the ERV for increased accuracy. The reference ERVs were identified from the long read-based genome assembly of the finch host species and included the sequences used to construct the ERV phylogeny. Non-reference ERVs that did not exist in the genome assembly were only detected with short reads, and therefore limited to sampling LTR sequence to assign them a reference ERV "tag". To assign each individual locus to the ERV phylogeny would require long-read sequencing and reconstruction of the provirus at each locus, which at this time is not possible. For now, assigning each locus a reference ERV as a "tag" based on sequence similarity of the LTR is sufficient for tracking ERV segregation in the populations we examined, which was our aim. Future investigations of the functional interaction between particular ERV loci and host species evolution would greatly benefit from recovering the full ERV sequence and a thorough systematics investigation, but in the context of this study the challenge of doing that for the thousands of non-reference ERVs we reported is not justified by the impact it would have on the results and conclusions we drew.

7) Interestingly, based on Fig 3, it is clear that cPa140 and cPa260 are phylogenetically very distinct, belonging to very different retroviral genera, and yet, an ERV could be assigned to be similar to both of

them at the same time... Examination of the supplementary table in more detail revealed many of these similar occurrences. I also found that >92% of the putative ERV-containing loci identified are less than 2 illumina-read long, being 300 bp in length or shorter (24,811 loci / 26,964 loci), and only <2% of the loci were $\geq 1,000$ bp long (459 loci / 26,964 loci). I'm not sure how many of these are empty integration sites in the reference genome or are just very short sequences identified as ERVs. Specifically, I'm unsure how an ERV that is not present in the reference genome could be confidently classified and identified based on a reference-based read mapping analysis (Lines 264-272). This raises a serious concern that many of the ERVs identified could be false positive, and may in part explain the observed ambiguous ERV group assignments. Please clarify and explain the ERV characterisation protocol in more details to justify that they are not false positive and could be classified / identified confidently.

The very small loci do, indeed, represent empty pre-integration sites in the genome assembly. These are identified by the RetroSeq software, as described in the methods (starting at line 291). Naturally, the ERVs and/or solo-LTRs found in these locations cannot be fully characterized (see point 6 above), but that does not imply that they are systematically false positives. We have modified the text describing the detection & assignment process used for improved clarity.

Lines 299-301:

The coordinates of each locus identified by RetroSeq correspond to the empty pre-integration site which is lacking an ERV insertion in the reference genome assembly.

ERV integration site distribution

8) Lines 129–134: These findings are rather interesting, but the authors hardly discuss the results. I would like to see the authors compare and contrast these findings against those of other ERVs, like human ERVs, and perhaps further discuss the nature of ERV integration / selection process.

We agree that there are some interesting comparative genomics questions that arise here, but a direct comparison is limited by two things. The first is that most genomes with well annotated ERVs are so distantly related that a lack of synteny would render chromosomal position less informative than we would hope. Secondly, we have limited power to determine the orientation of the non-reference ERV loci and therefore can't determine smaller scale relationships between ERV enrichment and particular genes.

ERV and host co-evolution

9) The authors report the host distributions of some ERVs in detail, e.g. those of cPa260 ERVs (Lines 135-149), cPa452 ERVs (Lines 150-157), and cPa363 ERVs (Lines 158-166). However, the discussions could be much richer, and evolutionarily more meaningful if they were done in the host evolutionary context, which is missing entirely from the work. One suggestion is to estimate the host phylogeny, and map the ERV abundances onto the host tree, or track the ERVs' radiation along the host speciation process to see how they co-diversify along each other, for example.

An additional panel has been added to figure 4 associating the host phylogeny with the distribution of cPa260. Our objective with this panel is to more explicitly demonstrate the association between the expansion of a particular *Beta-like-1* ERV and the speciation of the finches. Supplementary figure 3 associating the ERV enrichment with each species in the context of the host phylogeny has also been included.

Additional text has been added to the discussion to more closely connect the ERV and Darwin's Finch evolutionary history, beginning at line 211:

A conclusion from these analyses is that while *Beta-like-1*, *Beta-like-2*, and *Gamma* ERV clades saw large expansions in numbers throughout the Darwin's finch radiation, no single host species or monophyletic group of host species account for the overall increase in ERVs. With some notable exceptions discussed below, ERV enrichment does not follow an obvious pattern across the Darwin's finch radiation. This rules out the possibility that some extant descendants of the original colonists of the Galápagos islands were especially susceptible to retroviral infection and ERV accumulation. Instead, our observations are consistent with the standing ERV variation being shaped by infection bursts distributed across the entire Darwin's finch radiation followed by demographic effects such as repeated bottlenecks and gene-flow. The most abundant of these ERVs allow us to infer some of this history due to a large number of integrations and relatively high segregating frequency in some species.

This conclusion is drawn from the addition of figure 4b and Supplementary figure 3, the legends of which are:

Fig. 4 | Relative abundance of common ERVs in example species. ... b, Phylogeny of the non-hybrid Darwin's finch species with cPa260 MIR. Species in this phylogeny that also appear in panel **a** are highlighted with a border. Three of the four species enriched for cPa260 belong to the ground finch group, with the tree finch *C. parvulus* being the exception.

Supplementary Fig. 3 | ERV enrichment in host phylogenetic context. The enrichment of each of the top 10 most abundant ERVs is shown for the investigated non-hybrid species of Darwin's finch. An MIR value > 1 (yellow cell) indicates that an ERV is more abundant within a given species than the median of abundance of across all species. An MIR value < 1 (blue cell) indicates that a given species is deficient in a particular ERV. In general, the distribution of enrichment in the top 10 most abundant ERVs does not follow an obvious pattern, and the same holds true when all ERVs are considered, indicating that no single species or clade of Darwin's finches are relatively enriched for closely related ERVs.

We have also included new results from an ongoing analysis of ERV frequency in genomic regions which are strongly linked to small, medium, and large ground finch beak size phenotype. We describe them in the new Table 1 and the following text:

Results line 170:

Though we assume a neutral mode of evolution for the majority of identified ERVs, potential effects on host genome function cannot be excluded. Thus, we investigated ERV frequencies in 28 genomic regions showing strong genetic differentiation between small, medium, and large ground finches, previously identified by Rubin & Enbody *et al.*⁹ Within these regions, we identified 13 ERV loci at high frequencies in the large ground finch (*G. magnirostris*), and low frequencies in the small ground finch (*G. fuliginosa*), or *vice versa* (Table 1). The medium ground finch (*G. fortis*) was intermediate in frequency for these ERV loci.

Discussion line 255:

The potential role of selection in shaping ERV variation is currently under investigation. Allelic frequencies of the 13 ERV loci described in Table 1 covary with body and beak sizes across the Darwin's finch phylogeny⁹. While a causal role of an ERV locus in beak and/or body size

phenotypes remains to be confirmed, we conclude that at least part of the identified ERV variation described herein has been shaped by selection through hitchhiking on other causal variants. The outcome is ERVs with large frequency differences between closely related finch species at critical genomic locations.

The association of these ERVs with ground finch phenotype is under current investigation and will be described in a following manuscript.

10) Despite the name of the work “Spatiotemporal variations in retrovirus-host interactions among Darwin’s finches”, spatio- (Lines 167-177) and temporal- (Lines 202-224) variations of the ERVs are minimally investigated. In fact, direct temporal information of the ERVs is missing entirely from the work, and their potential temporal variations are only minimally discussed, inferred indirectly from varying ERV’s host distributions. Direct comparison of the ERV and host age estimates in the context of the virus and host co-phylogenies and spatial distributions could be highly informative and helpful with the discussion of the virus-host co-evolutionary process. Although the authors mention that “higher sequencing coverage would be required for a confident [age] estimate” (Lines 233-234); this should be possible for their top 10 ERVs at least.

We hope that the inclusion of the additional panel to figure 4 and the additional supplementary figure 3 more explicitly connects the expansion of ERV groups with the speciation history of the finches. The point made about age estimates is one that we will clarify in the text, but the issue here is that the detection of low frequency ERVs is challenging because they are almost exclusively present as heterozygotes. Our detection method, which relies on short reads, has difficulty identifying heterozygous non-reference ERV integrations when read coverage is low. Therefore, regardless of the abundance of a particular ERV, the low frequency sites will have enough of a false-negative detection rate in low-coverage samples to confound age estimates. The only way to achieve high confidence age estimates using the same or similar detection methods is to increase short read coverage.

Minor comments

1) Lines 264 and 273: I found the phrases “Non-assembly ERV insertion mapping” and “Assembly ERV insertion mapping” rather confusing. Please consider changing them to something else more reflective of the nature of the analysis aims.

The observed variation is identified either as present or missing in the assembly, and these analyses require different approaches 1. “Non-assembly ERV insertion mapping” identifying the loci present in the overall sample set, but missing from the assembly, and 2. “Assembly ERV insertion mapping” deals with calling absence of insertion, at ERV-loci present in the assembly, in a given individual of the population.

2) Fig 4 and Fig S3: Please sort their x axis to be in the same order.

We thank the reviewer for pointing this out and have corrected this.

3) The term ‘polymorphic’ in this work appears to specifically mean ‘insertionally polymorphic’, and

doesn't refer to any other kinds of polymorphisms, like nucleotide or structural polymorphisms. If so, please change the term throughout the text to be more specific and precise.

We thank the reviewer for pointing out this potentially confusing wording, and have specified 'insertionally polymorphic' at first use.

Line 82:

To identify insertionally polymorphic (hereafter referred to as polymorphic) ERVs in the finch populations.

Reviewer #2 (Remarks to the Author):

In this manuscript, Hill et al. explored the diversity and evolution of endogenous retroviruses (ERVs) in Darwin's finches that represent a text-book example of adaptive radiation. Briefly, the study investigated the diversity of ERVs in Darwin's finches, and reported the insertional polymorphism of ERVs across these finches. However, I don't see many interesting results expected when reading the title (for example, the relationship between the evolution of ERVs and the adaptive radiation of Darwin's finches). In my opinion, the analyses seems to be more or less superficial and might not represent the advances required for a Nature Communications paper.

Major comments:

1. Line 82-88. It remains unclear why a reference ERV mapping library should be reconstructed. Based on the context, I guess a reference ERV represents a virus that invaded the finch genomes.

The identification of an ERV using short reads is complicated by the fact that the only sequence information that is accessible, at the most, is one read length + the read pair insert size into the ERV insertion from either end. This region will be almost exclusively composed of an ERV's LTR region which is much less suitable for determining the phylogenetic relationship between ERVs than the protein coding *gag-pol-pro* genes of the provirus. Therefore, we first constructed a reference ERV mapping library based on ERVs identified in the host reference genome, which was assembled using long reads. Using that set of ERVs from the host genome we were able to construct a reliable phylogeny of these representative sequence using the informative protein coding regions. This set of ERVs with a well determined phylogenetic relationship then served as the reference library for the many non-reference ERVs which were identified in the sequenced samples. Since we are limited by sequence similarity of non-reference ERV LTRs, we certainly benefit from an established phylogeny of very closely related ERVs to assign the non-reference ERVs an identity. The reviewer is correct that all the cPa prefixed ERVs in the reference library were identified as ERVs in the reference sample and were therefore present in the genome assembly.

2. Line 89-92: The logic for mining ERV polymorphism is not well explained. Its performance appears to be obscure. I suggest an alternative pipeline that mined ERVs in each individual first and assigned orthologous relationships, which is much more straight-forward. Of course, there are other ways. Moreover, I suggest a figure showing ERV polymorphism across individuals could be provided.

We hope that the answer to the previous comment clarifies this issue. We agree with the reviewer that one possible strategy would be to directly compare orthology between all the loci. However, this is impossible with the data at hand since the majority of identified loci were not present in the reference sample, and therefore only short reads which probed the LTR regions of ERV insertions were available to provide identifying sequence. This LTR sequence was inadequate to identify orthology between loci, but was mappable to the previously constructed ERV library and thereby provide an identification.

3. The ERV polymorphism is not well connected with host population history. I guess the authors tried, but failed to figure out. Maybe simulation work will provide some insights.

We hope the addition of a new panel to figure 4 and the new supplementary figure 3, which more explicitly connects the expansion of ERV groups with the most recent phylogeny of

Darwin's finches, addresses this concern.

Additional text has been added to the discussion to more closely connect the ERV and Darwin's Finch evolutionary history, beginning at line 212:

A conclusion from these analyses is that while *Beta-like-1*, *Beta-like-2*, and *Gamma* ERV clades saw large expansions in numbers throughout the Darwin's finch radiation, no single host species or monophyletic group of host species account for the overall increase in ERVs. With some notable exceptions discussed below, ERV enrichment does not follow an obvious pattern across the Darwin's finch radiation. This rules out the possibility that some extant descendants of the original colonists of the Galápagos islands were especially susceptible to retroviral infection and ERV accumulation. Instead, our observations are consistent with the standing ERV variation being shaped by infection bursts distributed across the entire Darwin's finch radiation followed by demographic effects such as repeated bottlenecks and gene-flow. The most abundant of these ERVs allow us to infer some of this history due to a large number of integrations and relatively high segregating frequency in some species.

This conclusion is drawn from the addition of figure 4b and supplementary figure 3, the legends of which are:

Fig. 4 | Relative abundance of common ERVs in example species. ... **b**, Phylogeny of the non-hybrid Darwin's finch species with cPa260 MIR. Species in this phylogeny that also appear in panel **a** are highlighted with a border. Three of the four species enriched for cPa260 belong to the ground finch group, with the tree finch *C. parvulus* being the exception.

Supplementary Fig. 3 | ERV enrichment in host phylogenetic context. The enrichment of each of the top 10 most abundant ERVs is shown for the investigated non-hybrid species of Darwin's finch. An MIR value > 1 (yellow cell) indicates that an ERV is more abundant within a given species than the median of abundance of across all species. An MIR value < 1 (blue cell) indicates that a given species is deficient in a particular ERV. In general, the distribution of enrichment in the top 10 most abundant ERVs does not follow an obvious pattern, and the same holds true when all ERVs are considered, indicating that no single species or clade of Darwin's finches are relatively enriched for closely related ERVs.

We have also included new results from an ongoing analysis of ERV frequency in genomic regions which are strongly linked to small, medium, and large ground finch beak size phenotype. We describe them in the new Table 1 and the following text:

Results line 170:

Though we assume a neutral mode of evolution for the majority of identified ERVs, potential effects on host genome function cannot be excluded. Thus, we investigated ERV frequencies in 28 genomic regions showing strong genetic differentiation between small, medium, and large ground finches, previously identified by Rubin & Enbody *et al.*⁹ Within these regions, we identified 13 ERV loci at high frequencies in the large ground finch (*G. magnirostris*), and low frequencies in the small ground finch (*G. fuliginosa*), or *vice versa* (Table 1). The medium ground finch (*G. fortis*) was intermediate in frequency for these ERV loci.

Discussion line 255:

The potential role of selection in shaping ERV variation is currently under investigation. Allelic frequencies of the 13 ERV loci described in Table 1 covary with body and beak sizes across the

Darwin's finch phylogeny⁹. While a causal role of an ERV locus in beak and/or body size phenotypes remains to be confirmed, we conclude that at least part of the identified ERV variation described herein has been shaped by selection through hitchhiking on other causal variants. The outcome is ERVs with large frequency differences between closely related finch species at critical genomic locations.

The association of these ERVs with ground finch phenotype is under current investigation and will be described in a future manuscript.

Minor comments:

L80: the recently released assembly of the small tree finch should be introduced, because it is a highly contiguous assembly.

The assembly is presented in a separate work and the updated reference is presented in the text.

L95-96: These numbers are not equal to 26,962.

We thank the reviewer for identifying this error. We confirmed 26,964 loci to be correct, and updated the text accordingly. Two DELLY loci had mistakenly been excluded from the sum, but were present for all analyses.

L99-101: The meaning of these numbers is unclear. Please clarify.

We hope the above explanation on how the loci were identified as well as the relationship of the reference ERVs to the identified loci clarifies these numbers.

Reviewer #3 (Remarks to the Author):

NCOMMS-22-09493

This paper investigates the diversity and distribution of endogenous retroviruses (ERVs) in the genomes of Darwin's finches and close relatives. They leverage a large dataset of sequenced genomes from more than 20 species to characterize ERVs and document patterns of diversity and relics of host-viral interactions. The authors report a large diversity of ERVs among these species and highlight a number of interesting patterns. Few ERVs (< 2%) were found in all species, suggesting that the spread of ERVs is dynamic and much diversification has happened since the radiation of finches. They also draw attention to technical sources of variation, namely that there is a strong relationship between sequencing coverage and ERV diversity, and that the sample size of individuals sequenced from a given population also appears to affect ERV diversity. I think this is an interesting topic and there is a lot of promise in this dataset.

The authors promote the strength of the Darwin's finch system for studies of evolution. However, I think the manuscript is lacking integration of the ERV results with the biological context. I'm not clear about exactly why Darwin's finches are an important and revealing system for this study, or exactly what new conceptual ground these data cover. I didn't have a clear understanding from the introduction how the authors were planning to use their investigation of ERVs to solve outstanding questions about either finch evolution and/or host-pathogen coevolution. I understand that the authors may have prepared their manuscript with word limits in mind; however, I think that there are several areas where either expanded background or discussion would assist the reader's understanding of the importance of this study.

For example: The authors suggest that this screening of ERVs in the Darwin's finch radiation helps "evaluate aspects of the natural history of the entire Darwin's finches' radiation." I felt any mention of natural history in the discussion was lacking. Can the authors elaborate?

The authors investigate ERVs in both "pure" species as well as several hybrid lineages. Why were hybrids included in this study? What do ERVs reveal in those lines? On line 147 the authors note that hybrids of *Geospiza* finches have cPa260 at the same loci as the parental species, indicating that these ERVs are transmitted vertically. I would assume though that ERVs are heritable so I'm left wondering what is revealing about sequencing these hybrids?

Our rationale for choosing samples was simply to use everything that was available to us. The inclusion of hybrids was fortunate because they served as a validation of our detection methods, since we recover the patterns that inheritance would predict. In the case of the non-hybrid species we must infer what ERV loci extinct parental species may have carried based on the extant species. Fortunately, we had samples from the hybrid and parental species and when we examined the hybrids and found the high degree of overlap with the parental species, we confirmed that these loci were being inherited as expected, thus validating our detection method since either false positives or false negatives would have produced loci which were not shared between parental and hybrid species. Since loci in hybrid species were almost all accounted for in the parental species, our detection method must be accurate to a reasonable degree.

Line 255: The authors provide some tantalizing ideas of how ERVs could be used to further resolve the Darwin's finch phylogeny. I understand that was not part of the current study, but I think that

synthesis along just those lines would help integrate these new data within a broader context about ecology and/or evolution.

We absolutely agree that using the directionality of pre-integration site to provirus (ERV) to solo-LTR as markers to resolve the phylogeny is a compelling goal. One obstacle to that is the inability to distinguish between ERVs and solo-LTRs using the short read detection methods employed in this study. A study currently in progress in which we examine selected loci with long reads may overcome that problem and allow progress towards host phylogeny resolution.

ERVs are neat in studies of evolutionary biology because unlike point mutations they are unlikely to revert. Thus, the authors point out that two species that have a common ERV almost certainly inherited it from a common ancestor (i.e. convergence of ERVs is unlikely). However, with so many of the ERVs private to certain species I was left wondering how common loss of ERVs is. Can the authors provide any more information about how much to expect the loss of these variants to create the observed patterns and whether loss can ever be definitively inferred? For instance, the authors report that one of the outgroup birds, *L. noctis*, had a small proportion of cPa260, an ERV that is more common in Darwin's finches. The authors suggest that changes in viral activity post speciation could have produced the variation of cPa260 in the Darwin's finch radiation. Could it also be possible that selection or population bottlenecks in *L. noctis* could have reduced variation in that species?

That's a plausible alternative, however it's hard to imagine a bigger bottleneck than that experienced by the finches initially colonizing the Galápagos islands. That being said, demographic effects have surely played a role shaping current ERV frequencies in the island populations, but the rate of gene flow is still unknown. We hope that this and future work on ERV patterns in Darwin's finches will be useful to projects which aim to quantify these phenomena of demographic history.

How much of these patterns could be explained by demographic history? For instance, does population size affect ERV diversity? e.g. *G. fortis* on Santa Cruz or Isabela has a very large population, but would be substantially smaller on Pinta. Same thing *C. parvulus* on Santa Cruz vs. Santa Fe). Is ERV diversity correlated with population size or even something as crude as island size? How much of the ERV diversity should we expect to be a product of antagonistic co-evolution between hosts and pathogens vs. a product of neutral processes of drift etc?

Disentangling these important evolutionary forces that shaped the observed ERV distribution is as challenging as it is interesting. Our basic assumption is that the vast majority of ERV loci have evolved neutrally and so their distribution would be similar to that of any other neutral marker, for example a typical SNP. All of the demographic effects that would underlie the observed frequency distribution of a SNP should have the same effect on standing ERV variation. Despite trying, we could neither rule out nor confirm a differential retroviral infection history between the islands that may have contributed to the different ERV integration locus frequencies between island populations. Addressing this would require larger island sample sizes as well as a clearer understanding of rates of gene flow between islands. While we treat all loci as neutral, and we are certain that the vast majority do evolve neutrally or nearly-neutrally, we are currently studying the connection between a few selected ERV loci and phenotype in order to identify loci that might have contributed to speciation and appear in regions of strong purifying, positive, or balancing selection.

I think that integration of these results with the evolutionary history of this radiation is a bit lacking. For instance, we know from recent genomic work (Lamichhaney et al. 2015/16) that warbler finches diverged first in the Darwin's finch radiation and the Cocos finch emerged later and ultimately ended up outside the Galapagos on Cocos Island. Could these evolutionary relationships be used to infer something about the transmission patterns of ERVs?

We hope that the addition of a new panel to figure 4 and a new supplementary figure 3 draws a more explicit relationship between ERV distribution and host species phylogeny with the illustrated speciation history of the host species clade. While more precise application of ERVs to the task of phylogenetic resolution of the host species is a possibility for the near future, it relies upon the generation of whole genome long read sequencing data for more of the host species.

Minor comments:

Line 158: *C. olivacea* is highlighted here as an outlier finch that was enriched for cPa363. This species also seems to be lightly sampled ($N = 5$) compared to the rest of the species. Do the authors have any reason to believe that sampling artifacts could have produced this result?

This cannot be excluded. However, we must consider that a low sample number would have decreased the probability of sampling a locus which may have been segregating in the species but was not in one of the samples. For cPa363 to appear enriched as a result of a small sample size, every other ERV group would have had to be undersampled relative to the true population frequencies for that single group to appear overly abundant. Given the very large number of loci, this would be an exceedingly unlikely outcome.

Line 209: Differences in cPa260 ERVs in the *G. magnirostris* populations on Genovesa and Daphne populations: There was a large difference in sample sizes of these two populations (10 vs. 5). Could that explain some of the difference?

The small sample size on Daphne does increase the uncertainty, but, like above, the other ERV-groups are more consistent in the same comparison.

Fig 3b. Apologies, I'm not an expert in this field but I don't understand how to interpret this panel. Could the authors consider rephrasing or expanding the legend?

This panel's purpose is to illustrate the false negative detection rate of ERV loci that should be present as ERV insertions (fixed) in all samples. The rationale is that if an ERV insertion is present in at least one sample of every species then it must have been inherited from the common ancestor of all finches, and since an ERV insertion is very rarely completely lost then we would expect the insertion frequency of such an ERV to be 1.0 (fixed) in all populations. This panel legend has been expanded to hopefully improve clarity.

Fig. 3 | ERV phylogeny and heatmap. ... b, Frequency histogram of ERVs at loci that contain at least one ERV identification in all species. Loci that fit this criterion are assumed to pre-date Darwin's finch speciation and are therefore expected to be fixed in all populations. Observed

frequencies <1.0 of these ERVs, can be assumed to be the result of false negative identification calls.

Fig S3: Recommend adding genus abbreviations before species name (e.g. "G. fortis")

We thank the reviewer for the good suggestion. The Fig S4 (Previously S3), and Fig S6 (Previously S5) have been updated accordingly.

Reviewer #4 (Remarks to the Author):

The manuscript submitted by Hill et al describes a unique study that leverages finch genome data and recent finch genome assembly and the rich knowledge base centering on the natural history of Darwin's Finches in the Galápagos islands to explore the population genetics of endogenous retrovirus loci (ERVs) overlapping in time with the radiation of new species/lineages. While on the surface the study is mostly a descriptive cataloguing, it potentially provides a wealth of insight into virus-host interactions as they play out over generational timescales. That is to say, this manuscript could lay important groundwork for follow up studies of these data and for similar situations in nature where species history and corresponding sequence data for multiple closely allied species are available. One suspects that treating the individual ERV insertions as markers might ultimately provide insight into difficult-to-resolve host relationships. While much of that is beyond this first study, this paper lays the groundwork.

I have a couple of minor comments to aid in accessibility of the manuscript to the variety of readers who might be interested:

Some brief explanation of the "beta-like", "gamma-like", etc., terminology would be useful - even just a sentence with appropriate citations.

I encourage the authors to consider being clearer in distinguishing between "ERV" as a group, "ERV" as a specific insertion/locus.. For example, in the Discussion in lines 190 and 192, it would be more precise and less confusing to say "...homozygous, ERV loci..." rather than homozygous ERV, and similarly "heterozygous ERV loci" in line 192. This can also be an issue when discussing variation, frequency, abundance, etc., where it needs to be clear whether one is speaking of variation at a specific locus across a population versus variation within a group of ERV comprising many loci distributed across a genome.

ERV as loci associated by similarity to assembly ERVs in phylogenetic context. ERVs are part of the retrovirus evolutionary continuum but nomenclature and definitions are still debated and without making this priority for identifying segregating alleles in the populations, the assignments in this study indicate closest retroviral genera such as *betaretroviruses* and *gammaretroviruses* in the phylogeny.

Line 199-201 - does this imply that ongoing or recent infection by retroviruses closely overlapped in time with the arrival and early radiation of the finches? Seems like this might be worth more discussion. Also, might the authors include some speculation as to the possible impact of the viruses on evolution of the host lineages? It would seem that these data are well suited for asking questions about the potential contributions of ERV to genome evolution.

We agree that a more explicit connection between ERV distribution and host species phylogeny is warranted. We have expanded figure 4 to include a new panel highlighting the association between cPa260 and host evolutionary history, as well a new supplementary figure 3 for the top 10 ERVs. The conclusion we drew from this was that this is likely new activity among Darwin's finches around and after the time of colonization of Galápagos.

We have also included new results from an ongoing analysis of ERV frequency in genomic regions which are strongly linked to small, medium, and large ground finch beak size phenotype. We describe them in the new Table 1 and the following text:

Results line 170:

Though we assume a neutral mode of evolution for the majority of identified ERVs, potential effects on host genome function cannot be excluded. Thus, we investigated ERV frequencies in 28 genomic regions showing strong genetic differentiation between small, medium, and large ground finches, previously identified by Rubin & Enbody *et al.*⁹ Within these regions, we identified 13 ERV loci at high frequencies in the large ground finch (*G. magnirostris*), and low frequencies in the small ground finch (*G. fuliginosa*), or *vice versa* (Table 1). The medium ground finch (*G. fortis*) was intermediate in frequency for these ERV loci.

Discussion line 255:

The potential role of selection in shaping ERV variation is currently under investigation. Allelic frequencies of the 13 ERV loci described in Table 1 covary with body and beak sizes across the Darwin's finch phylogeny⁹. While a causal role of an ERV locus in beak and/or body size phenotypes remains to be confirmed, we conclude that at least part of the identified ERV variation described herein has been shaped by selection through hitchhiking on other causal variants. The outcome is ERVs with large frequency differences between closely related finch species at critical genomic locations.

The association of these ERVs with ground finch phenotype is under current investigation and will be described in a future manuscript.

Reviewers' Comments:

Reviewer #1:

Remarks to the Author:

The authors have satisfactorily addressed all of my concerns.

Reviewer #2:

Remarks to the Author:

I think the newly added lines 186-193 cannot establish a plausible relationship between ERVs and functions.

Reviewer #3:

Remarks to the Author:

The authors have revised their manuscript and responded in detail to the comments by myself and the other three reviewers of the first submission. I see that they have done a thorough investigation of the available sequence data and the diversity of ERVs that they report is interesting.

However, I still have two primary concerns that were raised by myself and others in the initial review that I do not believe have been adequately addressed.

First, the revision adds little to better place these data within either a spatial or temporal context. There is little to no discussion of spatial variation in ERVs. Temporally, the authors add a panel to figure 4 with the Darwin's finch phylogeny, where each species is labeled by a number that indicates the relative abundance of the most variable ERV. Although the reader can see that the number varies among species, and that there are some visible patterns (Geospiza finches tend to have higher MIR), there is no explicit phylogenetic analysis of this ERV, and no biological reason to focus on this particular ERV, besides the fact that it is common. The authors also add a similar supplementary figure where they correlate the 10 most common ERVs with the Darwin's finch phylogeny and color the ERVs based on whether they are more or less common in that species. Why make MIR binary when it's shown as a continuous variable in other figures? Neither of these additions feels revelatory. In the absence of a clear trend, or an explicit analysis we're left concluding that there is not particular association of ERVs with phylogeny. With that conclusion in mind, in the revised manuscript it is still unclear what ERVs reveal about their hosts' evolution.

In addition, the analyses and discussion added to integrate the results within a phylogenetic and spatial context seem cursory. For example: Line 281: "Allelic frequencies of the 13 ERV loci in Table 1 covary with beak and body sizes across the Darwin's finch phylogeny." These loci are only correlated with small, medium and large ground finches, not across the phylogeny. I think it's fine to highlight that some ERV were associated with loci connected to phenotype in the ground finches, but the language should be precise.

Overall, in the revision the authors have only modified their discussion by adding two paragraphs. Some of the language of these additions is not well supported by the data.

E.g. Line 284: "we conclude that at least part of the identified ERV variation described herein has been shaped by selection through hitchhiking on other causal variants." I do not understand where this conclusion has come from. The authors pick (somehow) 13 ERV loci whose frequency is correlated with the small-medium-large ground finch continuum. Could this not be due to chance? How many ERVs were not correlated with this continuum among the 28 regions studied?

Second, I'm still concerned that many of the patterns reported are driven by sample sizes. Although it is not easy to glean total sample sizes from Figure 1, it also appears that the species with the highest MIR for cPa260 also tend to be well sampled.

The within-species variation in ERVs illustrated in Figures 4A and S4 emphasizes my concerns about sample size. I'm concerned that many of the patterns the authors detected are artifacts of sequencing so few samples, and that their inferences about differences in MIR between species would change with more samples. Furthermore, there is no formal comparison of within-species variation among island populations so it is not clear how much population structure may influence results.

"Line 245: the variation between the warbler-, tree- and ground finch groups in, for example, cPa452 ERV distribution along the genome indicates temporal variations in host tolerance to ERV accumulation during the Darwin's finch radiation."

I'm not sure the data support this assertion. Rather than varying tolerance to ERV accumulation over time, is it not also possible that ancestors of certain clades were more exposed to ERVs than others and/or that certain ERVs were lost by chance? This statement seems to contradict the preceding paragraph in yellow that emphasizes demographic processes in shaping ERV patterns in the radiation.

Fig 4b. For clarity, I recommend that the legend specify that the MIR is in parentheses.

Reviewer #4:

Remarks to the Author:

My original comments were minor and largely aimed at accessibility and clarity of this (somewhat) confusing manuscript. Overall, a very nice study, and I'm satisfied that the authors have responded appropriately to my concerns.

We thank you for the additional comments to improve our revised manuscript. We have responded to the comments point-by-point below, and updated the manuscript accordingly.

Patric Jern

Reviewer #1 (Remarks to the Author):

The authors have satisfactorily addressed all of my concerns.

We thank the reviewer for comments and suggestions.

Reviewer #2 (Remarks to the Author):

I think the newly added lines 186-193 cannot establish a plausible relationship between ERVs and functions.

We agree that the data available to us presents limitations for confident connection between ERVs and host function, which is the scope for an initiated study. We further revised the text: “Though we assume a neutral mode of evolution for the majority of identified ERVs, potential effects on host genome function cannot be established from the currently available data”, and “This segregation reflects the observed patterns in ground finches described by Rubin *et al.*⁹, which indicates that low frequency ERV variants were part of the selected haplotype”.

Reviewer #3 (Remarks to the Author):

The authors have revised their manuscript and responded in detail to the comments by myself and the other three reviewers of the first submission. I see that they have done a thorough investigation of the available sequence data and the diversity of ERVs that they report is interesting.

However, I still have two primary concerns that were raised by myself and others in the initial review that I do not believe have been adequately addressed.

First, the revision adds little to better place these data within either a spatial or temporal context. There is little to no discussion of spatial variation in ERVs. Temporally, the authors add a panel to figure 4 with the Darwin’s finch phylogeny, where each species is labeled by a number that indicates the relative abundance of the most variable ERV. Although the reader can see that the number varies among species, and that there are some visible patterns (Geospiza finches tend to have higher MIR), there is no explicit phylogenetic analysis of this ERV, and no biological reason to focus on this particular ERV, besides the fact that it is common.

The cPa260 was selected as an example because it is abundant in some species, but also rare in other species. As described in the text and illustrated in the figures, each of the ERVs we identified arrived at their observed frequencies through a unique history of evolution. We made the data frame and analysis scripts available to enable future analyses regarding population frequencies of any of the other ERVs. We focus on providing a broad description of the dynamics of the entire ERV landscape and the variation between species in an adaptive radiation, and do not attempt to reconstruct the detailed history of a particular locus or ERV group in this study. Thus, and as previously responded to the other reviewers, further high coverage long read sequencing of these precious samples for phylogenetic comparison of all ERV loci presents scope for a future study.

The authors also add a similar supplementary figure where they correlate the 10 most common ERVs with the Darwin's finch phylogeny and color the ERVs based on whether they are more or less common in that species. Why make MIR binary when it's shown as a continuous variable in other figures? Neither of these additions feels revelatory. In the absence of a clear trend, or an explicit analysis we're left concluding that there is not particular association of ERVs with phylogeny. With that conclusion in mind, in the revised manuscript it is still unclear what ERVs reveal about their hosts' evolution.

This is a reasonable conclusion and also important. As stated in the manuscript we find it valuable to know that there is no evidence for a massive expansion of ERVs throughout any particular clade. Scenarios that would have resulted in a "particular association of ERVs with phylogeny" can now be deemed less likely, such as clade specific rates of retroviral infection, loss of ERVs from the population, etc.

In addition, the analyses and discussion added to integrate the results within a phylogenetic and spatial context seem cursory. For example: Line 281: "Allelic frequencies of the 13 ERV loci in Table 1 covary with beak and body sizes across the Darwin's finch phylogeny." These loci are only correlated with small, medium and large ground finches, not across the phylogeny. I think it's fine to highlight that some ERV were associated with loci connected to phenotype in the ground finches, but the language should be precise.

This is a good point, and we have updated the text to reflect the focus on ground finches: "Allele frequencies at the 13 ERV loci described in Table 1 covary with body and beak sizes among the ground finches⁹."

Overall, in the revision the authors have only modified their discussion by adding two paragraphs. Some of the language of these additions is not well supported by the data.

E.g. Line 284: "we conclude that at least part of the identified ERV variation described herein has been shaped by selection through hitchhiking on other causal variants." I do not understand where this conclusion has come from. The authors pick (somehow) 13 ERV loci whose frequency is correlated with the small-medium-large ground finch continuum. Could this not be due to chance? How many ERVs were not correlated with this continuum among the 28 regions studied?

The reviewer makes a good point that we did not state how many ERV loci were detected in the 28 selective sweep regions. We found 379 ERV loci within the genomic regions associated with beak phenotype among the ground finches, of which 45 loci were identified at intermediate frequencies in *G. fortis*. Among those loci, 13 were carried in the 10% greatest absolute difference in frequency between *G. fuliginosa* and *G. magnirostris*. This selection maximized chance for identifying ERVs that were present from the establishment of the haplotype associated with beak phenotype, and either hitchhiked due to linkage with the causative variants, or even were perhaps causative themselves. Confident connection between these ERVs and their genomic contributions requires additional experimental verification in a future study. See also response to reviewer 2.

Second, I'm still concerned that many of the patterns reported are driven by sample sizes. Although it is not easy to glean total sample sizes from Figure 1, it also appears that the species with the highest MIR for cPa260 also tend to be well sampled.

The within-species variation in ERVs illustrated in Figures 4A and S4 emphasizes my concerns about

sample size. I'm concerned that many of the patterns the authors detected are artifacts of sequencing so few samples, and that their inferences about differences in MIR between species would change with more samples. Furthermore, there is no formal comparison of within-species variation among island populations so it is not clear how much population structure may influence results.

This pattern is coincidental. The way MIR was constructed as a comparison metric of ERV abundance was done to normalize for detection rate between populations and control for sample size and sequencing coverage disparity. Since MIR is a relative abundance measurement, scored for each individual, increasing sample size would not influence the MIR value of any particular ERV group. Thus, cPa260 was selected as an example due to its variable, rather than high, MIR (see response above).

"Line 245: the variation between the warbler-, tree- and ground finch groups in, for example, cPa452 ERV distribution along the genome indicates temporal variations in host tolerance to ERV accumulation during the Darwin's finch radiation."

I'm not sure the data support this assertion. Rather than varying tolerance to ERV accumulation over time, is it not also possible that ancestors of certain clades were more exposed to ERVs than others and/or that certain ERVs were lost by chance? This statement seems to contradict the preceding paragraph in yellow that emphasizes demographic processes in shaping ERV patterns in the radiation.

It is highly probable that many ERVs were lost to chance, and that neutral processes have shaped most of the currently observed ERV frequency distribution. We establish this earlier in the discussion and then point out particular cases (i.e. cPa452) which seem to violate this assumption. While determining the relative contribution of these evolutionary forces in the history of the finches is interesting, we must wait until more data is available to do more than speculate.

Fig 4b. For clarity, I recommend that the legend specify that the MIR is in parentheses.

Done: "Phylogeny of the non-hybrid Darwin's finch species with cPa260 MIR in parenthesis next to species abbreviations."

Reviewer #4 (Remarks to the Author):

My original comments were minor and largely aimed at accessibility and clarity of this (somewhat) confusing manuscript. Overall, a very nice study, and I'm satisfied that the authors have responded appropriately to my concerns.

We appreciate these positive comments.